# Genetic and environmental influences on adult human height across birth cohorts from 1886 to 1994

Aline Jelenkovic[1,2]*, Yoon-Mi Hur[3], Reijo Sund[1], Yoshie Yokoyama[4], Sisira H Siribaddana[5,6], Matthew Hotopf[7], Athula Sumathipala[5,8], Fruhling Rijsdijk[9], Qihua Tan[10], Dongfeng Zhang[11], Zengchang Pang[12], Sari Aaltonen[1,13], Kauko Heikkilä[13], Sevgi Y Öncel[14], Fazil Aliev[15,16,17], Esther Rebato[2], Adam D Tarnoki[18,19], David L Tarnoki[18,19], Kaare Christensen[20,21,22,23], Axel Skytthe[20,21], Kirsten O Kyvik[24,25], Judy L Silberg[26], Lindon J Eaves[26], Hermine H Maes[27], Tessa L Cutler[28], John L Hopper[28,29], Juan R Ordoñana[30,31], Juan F Sánchez-Romera[31,32], Lucia Colodro-Conde[30,33], Wendy Cozen[34,35], Amie E Hwang[34], Thomas M Mack[34,35], Joohon Sung[29,36], Yun-Mi Song[37], Sarah Yang[29,36], Kayoung Lee[38], Carol E Franz[39], William S Kremen[39,40], Michael J Lyons[41], Andreas Busjahn[42], Tracy L Nelson[43], Keith E Whitfield[44], Christian Kandler[45], Kerry L Jang[46], Margaret Gatz[47,48], David A Butler[49], Maria A Stazi[50], Corrado Fagnani[50], Cristina D'Ippolito[50], Glen E Duncan[51], Dedra Buchwald[52], Catherine A Derom[53,54], Robert F Vlietinck[53], Ruth JF Loos[55,56], Nicholas G Martin[57], Sarah E Medland[57], Grant W Montgomery[58], Hoe-Uk Jeong[59], Gary E Swan[60], Ruth Krasnow[61], Patrik KE Magnusson[48], Nancy L Pedersen[48], Anna K Dahl-Aslan[48,62], Tom A McAdams[9], Thalia C Eley[9], Alice M Gregory[63], Per Tynelius[64], Laura A Baker[47], Catherine Tuvblad[47,65], Gombojav Bayasgalan[66], Danshiitsoodol Narandalai[66,67], Paul Lichtenstein[48], Timothy D Spector[68], Massimo Mangino[68], Genevieve Lachance[68], Meike Bartels[69], Toos CEM van Beijsterveldt[69], Gonneke Willemsen[69], S Alexandra Burt[70], Kelly L Klump[70], Jennifer R Harris[71], Ingunn Brandt[71], Thomas Sevenius Nilsen[71], Robert F Krueger[72], Matt McGue[72], Shandell Pahlen[72], Robin P Corley[73], Jacob v B Hjelmborg[20,21], Jack H Goldberg[74], Yoshinori Iwatani[75], Mikio Watanabe[75], Chika Honda[75], Fujio Inui[75,76], Finn Rasmussen[64], Brooke M Huibregtse[73], Dorret I Boomsma[69], Thorkild I A Sørensen[77,78,79], Jaakko Kaprio[13,80], Karri Silventoinen[1,75]

[1]Department of Social Research, University of Helsinki, Helsinki, Finland; [2]Department of Genetics, Physical Anthropology and Animal Physiology, University of the Basque Country, Leioa, Spain; [3]Department of Education, Mokpo National University, Jeonnam, South Korea; [4]Department of Public Health Nursing, Osaka City University, Osaka, Japan; [5]Institute of Research & Development, Battaramulla, Sri Lanka; [6]Faculty of Medicine & Allied Sciences, Rajarata University of Sri Lanka, Saliyapura, Sri Lanka; [7]NIHR Mental Health Biomedical Research Centre, South London and Maudsley NHS Foundation Trust and, Institute of Psychiatry Psychology and Neuroscience, King's College London, London, United Kingdom; [8]Research Institute for Primary Care and Health Sciences, School for Primary Care Research, Faculty of Health, Keele University, Staffordshire, United Kingdom; [9]MRC Social, Genetic & Developmental Psychiatry Centre, Institute of Psychiatry, Psychology & Neuroscience, King's College London, London, United Kingdom;

*For correspondence: aline. jelenkovic@helsinki.fi

Competing interests: The authors declare that no competing interests exist.

[10]Epidemiology, Biostatistics and Biodemography, Institute of Public Health, University of Southern Denmark, Odense, Denmark; [11]Department of Public Health, Qingdao University Medical College, Qingdao, China; [12]Department of Noncommunicable Diseases Prevention, Qingdao Centers for Disease Control and Prevention, Qingdao, China; [13]Department of Public Health, University of Helsinki, Helsinki, Finland; [14]Department of Statistics, Faculty of Arts and Sciences, Kirikkale University, Kirikkale, Turkey; [15]Faculty of Business, Karabuk University, Karabuk, Turkey; [16]Department of Psychology, Virginia Commonwealth University, Richmond, United States; [17]Department of African American Studies, Virginia Commonwealth University, Richmond, United States; [18]Department of Radiology and Oncotherapy, Semmelweis University, Budapest, Hungary; [19]Hungarian Twin Registry, Budapest, Hungary; [20]The Danish Twin Registry, University of Southern Denmark, Odense, Denmark; [21]Department of Public Health, Epidemiology, Biostatistics & Biodemography, University of Southern Denmark, Odense, Denmark; [22]Department of Clinical Biochemistry and Pharmacology, Odense University Hospital, Odense, Denmark; [23]Department of Clinical Genetics, Odense University Hospital, Odense, Denmark; [24]Department of Clinical Research, University of Southern Denmark, Odense, Denmark; [25]Odense Patient data Explorative Network (OPEN), Odense University Hospital, Odense, Denmark; [26]Department of Human and Molecular Genetics, Virginia Institute for Psychiatric and Behavioral Genetics, Virginia Commonwealth University, Richmond, Virginia, United States; [27]Department of Human and Molecular Genetics, Psychiatry & Massey Cancer Center, Virginia Commonwealth University, Richmond, Virginia, United States; [28]The Australian Twin Registry, Centre for Epidemiology and Biostatistics, The University of Melbourne, Melbourne, Australia; [29]Department of Epidemiology, School of Public Health, Seoul National University, Seoul, Korea; [30]Department of Human Anatomy and Psychobiology, University of Murcia, Murcia, Spain; [31]IMIB-Arrixaca, Murcia, Spain; [32]Department of Developmental and Educational Psychology, University of Murcia, Murcia, Spain; [33]QIMR Berghofer Medical Research Institute, Brisbane, Australia; [34]Department of Preventive Medicine, Keck School of Medicine of USC, University of Southern California, Los Angeles, United States; [35]USC Norris Comprehensive Cancer Center, Los Angeles, United States; [36]Institute of Health and Environment, Seoul National University, Seoul, South-Korea; [37]Department of Family Medicine, Samsung Medical Center, Sungkyunkwan University School of Medicine, Seoul, South-Korea; [38]Department of Family Medicine, Busan Paik Hospital, Inje University College of Medicine, Busan, Korea; [39]Department of Psychiatry, University of California, San Diego, San Diego, United States; [40]VA San Diego Center of Excellence for Stress and Mental Health, La Jolla, CA, United States; [41]Department of Psychology, Boston University, Boston, United States; [42]HealthTwiSt GmbH, Berlin, Germany; [43]Department of Health and Exercise Sciences and Colorado School of Public Health, Colorado State University, Colorado, United States; [44]Psychology and Neuroscience, Duke University, Durham, United States; [45]Department of Psychology, Bielefeld University, Bielefeld, Germany; [46]Department of Psychiatry, University of British Columbia, Vancouver, Canada; [47]Department of Psychology, University of Southern California, Los Angeles, United States; [48]Department of Medical Epidemiology and Biostatistics, Karolinska Institutet, Stockholm, Sweden; [49]Health and Medicine Division, The National Academies of Sciences, Engineering, and Medicine, Washington, United States; [50]Istituto Superiore di Sanità - National Center for Epidemiology, Surveillance and Health Promotion, Rome, Italy; [51]Washington State Twin Registry, Washington State University - Health Sciences Spokane, Spokane, United States; [52]Washington State

Twin Registry, Washington State University, Seattle, United States; [53]Centre of Human Genetics, University Hospitals Leuven, Leuven, Belgium; [54]Department of Obstetrics and Gynaecology, Ghent University Hospitals, Ghent, Belgium; [55]The Charles Bronfman Institute for Personalized Medicine, Icahn School of Medicine at Mount Sinai, New York, United States; [56]The Mindich Child Health and Development Institute, Icahn School of Medicine at Mount Sinai, New York, United States; [57]Genetic Epidemiology Department, QIMR Berghofer Medical Research Institute, Brisbane, Australia; [58]Molecular Epidemiology Department, QIMR Berghofer Medical Research Institute, Brisbane, Australia; [59]Department of Education, Mokpo National University, Jeonnam, South Korea; [60]Stanford Prevention Research Center, Department of Medicine, Stanford University School of Medicine, Stanford, United States; [61]Center for Health Sciences, SRI International, Menlo Park, United States; [62]Institute of Gerontology and Aging Research Network – Jönköping (ARN-J), School of Health and Welfare, Jönköping University, Jönköping, Sweden; [63]Department of Psychology, Goldsmiths, University of London, London, United Kingdom; [64]Department of Public Health Sciences, Karolinska Institutet, Stockholm, Sweden; [65]School of Law, Psychology and Social Work, Örebro University, Örebro, Sweden; [66]Healthy Twin Association of Mongolia, Ulaanbaatar, Mongolia; [67]Graduate School of Biomedical and Health Sciences, Hiroshima University, Hiroshima, Japan; [68]Department of Twin Research and Genetic Epidemiology, King's College, London, United Kingdom; [69]Department of Biological Psychology, VU University Amsterdam, Amsterdam, Netherlands; [70]Michigan State University, East Lansing, Michigan, United States; [71]Norwegian Institute of Public Health, Oslo, Norway; [72]Department of Psychology, University of Minnesota, Minneapolis, United States; [73]Institute for Behavioral Genetics, University of Colorado, Boulder, United States; [74]Department of Epidemiology, School of Public Health, University of Washington, Seattle, United States; [75]Osaka University Graduate School of Medicine, Osaka University, Osaka, Japan; [76]Faculty of Health Science, Kio University, Nara, Japan; [77]Novo Nordisk Foundation Centre for Basic Metabolic Research (Section on Metabolic Genetics), University of Copenhagen, Copenhagen, Denmark; [78]Department of Public Health, Faculty of Health and Medical Sciences, University of Copenhagen, Copenhagen, Denmark; [79]Institute of Preventive Medicine, Bispebjerg and Frederiksberg Hospitals, Copenhagen, Denmark; [80]Institute for Molecular Medicine FIMM, Helsinki, Finland

**Abstract** Human height variation is determined by genetic and environmental factors, but it remains unclear whether their influences differ across birth-year cohorts. We conducted an individual-based pooled analysis of 40 twin cohorts including 143,390 complete twin pairs born 1886–1994. Although genetic variance showed a generally increasing trend across the birth-year cohorts, heritability estimates (0.69-0.84 in men and 0.53-0.78 in women) did not present any clear pattern of secular changes. Comparing geographic-cultural regions (Europe, North America and Australia, and East Asia), total height variance was greatest in North America and Australia and lowest in East Asia, but no clear pattern in the heritability estimates across the birth-year cohorts emerged. Our findings do not support the hypothesis that heritability of height is lower in populations with low living standards than in affluent populations, nor that heritability of height will increase within a population as living standards improve.

## Introduction

Height is a classic anthropometric quantitative trait in humans due to its ease of measurement, approximately normal distribution and relative stability in adulthood. Since the studies of height in the late 19[th] and early 20[th] centuries (*Galton, 1886*; *Pearson and Lee, 1903*; *Fisher, 1919*), twin, adoption and family studies have shown that height is one of the most heritable human quantitative phenotypes (*Silventoinen, 2003*). More recently, genetic linkage studies have helped to elucidate the location of genetic effects in the genome (*Perola et al., 2007*) and genome-wide association (GWA) studies allowed identification of loci consistently associated with height in populations of different ancestry (*Cho et al., 2009*; *Hao et al., 2013*; *Lango Allen et al., 2010*; *N'Diaye et al., 2011*; *Wood et al., 2014*). Besides the genetic factors, a multitude of environmental factors, such as nutrition and childhood diseases, operate during the growth period and can affect the final attained height. These and other proximate biological determinants of height are further associated with social and economic conditions, which in turn are associated with living standards (*Bozzoli et al., 2009*; *Bogin, 2001*; *Eveleth and Tanner, 1990*; *Steckel, 2009*). The secular trend of increasing height over the 20[th] century observed in many parts of the world, which has slowed or stopped in most northern European countries, probably reflects the continuous improvement in the standard of living (*Eveleth and Tanner, 1990*; *Cole, 2003*; *Stulp and Barrett, 2016*). A recent study showed that the height difference between the tallest and shortest populations a century ago (19–20 cm) has remained the same for women and increased for men (*NCD Risk Factor Collaboration (NCD-RisC), 2016*) .

Twin and family studies have consistently estimated that the proportion of variation in adult height explained by genetic differences between individuals, or heritability, in general populations is approximately 0.80 (*Fisher, 1919*; *Silventoinen et al., 2003*; *Stunkard et al., 1986*). There is a hypothesis that heritability is not constant and can differ in environments having different amount of environmental variation. Accordingly, it has been suggested that heritability of height is lower in populations with low living standards compared with affluent populations since poverty can lead to a lack of basic necessities important for human growth in part of the population (*Steckel, 2009*). However, there is little direct evidence on this issue. A study in Finnish twins born between 1900 and 1957 showed that the heritability of height increased across birth cohorts born in the first half of the century when the standard of living increased and leveled off after World War II thus supporting this hypothesis (*Silventoinen et al., 2000*). Because this result needs to be replicated, we conducted an individual-based analysis of 40 twin cohorts from 20 countries. We aimed to analyze (i) the genetic and environmental contribution to the variation of adult height across nine birth-year cohorts covering more than 100 years and (ii) to assess whether the pattern varies by geographic-cultural region (Europe, North America and Australia, and East Asia).

## Results

In the pooled data (all twin cohorts together), mean height was greater in men than in women and increased over the birth-year cohorts in both sexes; the decrease ( > 1 cm) observed in the latest birth cohort mainly reflects differences in the distribution of different twin cohorts within each group (*Table 1*). Both means and variances were significantly different between twin cohorts in all birth-year and sex groups. Mean height was shorter in East Asia than in Europe and North America and Australia in all birth-year and sex groups. The increase in mean height over the birth cohorts (from 1940–1949 to 1980–1994) was substantially greater in East Asia than in the other two geographic-cultural regions. The variance of height was generally greater in men than in women, lowest in East Asia and greatest in North America and Australia, and showed a general trend to increase over the birth cohorts.

The variance of adult height explained by additive genetic, shared environmental and unique environmental factors by birth-year cohorts is presented in *Figure 1* (estimates with 95% confidence intervals (CIs) are available in *Supplementary file 1A*). In men, there was a trend for an increasing total variance from birth cohort 1940–1949 onwards; genetic variance also increased during this period but especially in the two latest birth-year cohorts (1970–1979 and 1980–1994). Height variance due to the environment shared by co-twins was significant from birth cohorts 1920–1929 to 1970–1979, being greatest from 1950 to 1969. The effect of environmental factors unique to each

**Table 1.** Descriptive statistics of age and height by sex, birth year and geographic-cultural region. Names list of the participating twin cohorts in this study: two cohorts from Australia (Australian Twin Registry and Queensland Twin Register), six cohorts from East-Asia (Korean Twin-Family Register, Mongolian Twin Registry, Osaka University Aged Twin Registry, South Korea Twin Registry, Qingdao Twin Registry of Adults and West Japan Twins and Higher Order Multiple Births Registry), 18 cohorts from Europe (Adult Netherlands Twin Registry, Berlin Twin Register, Bielefeld Longitudinal Study of Adult Twins, Danish Twin Cohort, East Flanders Prospective Twin Survey, Finnish Older Twin Cohort, FinnTwin12, FinnTwin16, Genesis 12–19 Study, Hungarian Twin Registry, Italian Twin Registry, Murcia Twin Registry, Norwegian Twin Registry, Swedish Twin Cohorts, Swedish Young Male Twins Study of Adults, TCHAD-study, TwinsUK and Young Netherlands Twin Registry), two cohorts from South-Asia and Middle-East (Sri Lanka Twin Registry and Turkish Twin Study) and 12 cohorts from North-America (California Twin Program, Carolina African American Twin Study of Aging, Colorado Twin Registry, Michigan State University Twin Registry, Mid Atlantic Twin Registry, Minnesota Twin Registry, NAS-NRC Twin Registry, SRI-international, University of British Columbia Twin Project, University of Southern California Twin Study, University of Washington Twin Registry and Vietnam Era Twin Study of Aging).

| | Age | | | Height | | | Europe | | | NA and Australia | | | East Asia | | |
| | All cohorts | | | All cohorts | | | | | | | | | | | |
| Birth year | Mean | SD | Range | N | Mean (F, p-value)[*] | SD (F, p-value)[†] | N | Mean | SD | N | Mean | SD | N | Mean | SD |
|---|---|---|---|---|---|---|---|---|---|---|---|---|---|---|---|
| Men | | | | | | | | | | | | | | | |
| 1886–1909 | 67.0 | 7.5 | 53.5–99.2 | 3747 | 171.6 (15, < 0.001) | 6.34 (2.5,0.019) | 3569 | 171.5 | 6.27 | 178 | 174.6 | 6.88 | | | |
| 1910–1919 | 52.2 | 16.2 | 20.0–95.8 | 9171 | 174.2 (23, < 0.001) | 6.72 (5.0,<0.001) | 4117 | 173.3 | 6.37 | 5052 | 174.9 | 6.91 | | | |
| 1920–1929 | 51.6 | 16.1 | 20.0–90.9 | 23147 | 175.4 (62, < 0.001) | 6.81 (5.7,<0.001) | 6382 | 173.9 | 6.42 | 16714 | 176.0 | 6.82 | | | |
| 1930–1939 | 57.5 | 10.5 | 33.5–83.2 | 12028 | 175.7 (413, < 0.001) | 6.70 (2.9,<0.001) | 9308 | 175.2 | 6.42 | 2658 | 178.1 | 6.78 | | | |
| 1940–1949 | 49.3 | 10.6 | 23.5–73.9 | 22967 | 177.4 (72, < 0.001) | 6.73 (2.5,<0.001) | 16629 | 177.0 | 6.53 | 6235 | 178.4 | 6.95 | 68 | 164.8 | 6.57 |
| 1950–1959 | 41.4 | 10.0 | 19.5–65.0 | 24560 | 178.4 (120, < 0.001) | 6.96 (6.5,<0.001) | 15199 | 178.5 | 6.73 | 9124 | 178.7 | 7.04 | 161 | 167.1 | 4.79 |
| 1960–1969 | 35.5 | 7.1 | 19.5–54.0 | 13264 | 179.0 (99, < 0.001) | 7.49 (2.3,<0.001) | 6218 | 179.6 | 7.04 | 6574 | 179.2 | 7.22 | 298 | 168.1 | 6.24 |
| 1970–1979 | 28.7 | 5.4 | 19.5–44.0 | 14975 | 179.9 (121, < 0.001) | 7.55 (5.5,<0.001) | 10339 | 180.7 | 7.01 | 3906 | 179.7 | 7.51 | 456 | 170.1 | 5.68 |
| 1980–1994 | 23.1 | 3.2 | 19.5–34.4 | 9948 | 178.4 (70, < 0.001) | 7.59 (4.9,<0.001) | 5077 | 178.8 | 7.22 | 4066 | 179.4 | 7.49 | 329 | 173.1 | 6.37 |
| Women | | | | | | | | | | | | | | | |
| 1886–1909 | 68.5 | 8.1 | 53.5–98.0 | 5423 | 160.2 (23, < 0.001) | 6.14 (3.3,0.006) | 5011 | 160.2 | 6.11 | 412 | 160.2 | 6.41 | | | |
| 1910–1919 | 62.0 | 10.9 | 43.6–95.9 | 7169 | 161.1 (18, < 0.001) | 5.93 (2.5,0.002) | 5621 | 161.0 | 5.85 | 1548 | 161.2 | 6.20 | | | |
| 1920–1929 | 59.7 | 11.4 | 37.5–91.7 | 10975 | 162.1 (65, < 0.001) | 5.99 (3.8,<0.001) | 7908 | 162.0 | 5.89 | 3052 | 162.4 | 6.16 | | | |
| 1930–1939 | 57.9 | 10.0 | 33.5–83.0 | 14610 | 162.7 (249, < 0.001) | 6.05 (5.8,<0.001) | 11226 | 162.5 | 5.83 | 3344 | 163.2 | 6.49 | | | |
| 1940–1949 | 49.9 | 10.2 | 23.5–74.0 | 28537 | 163.7 (175, < 0.001) | 6.19 (10.3,<0.001) | 20097 | 163.9 | 5.93 | 8285 | 163.5 | 6.57 | 100 | 153.6 | 5.33 |
| 1950–1959 | 41.3 | 9.5 | 19.5–64.0 | 31250 | 164.4 (146, < 0.001) | 6.58 (13.6,<0.001) | 18817 | 164.8 | 6.22 | 12080 | 164.1 | 6.78 | 225 | 155.1 | 5.10 |
| 1960–1969 | 35.8 | 6.9 | 19.5–54.3 | 20422 | 165.1 (163, < 0.001) | 7.00 (8.6,<0.001) | 9604 | 166.2 | 6.58 | 10182 | 164.6 | 6.87 | 438 | 156.8 | 5.17 |
| 1970–1979 | 29.3 | 5.4 | 19.5–44.3 | 19893 | 165.9 (180, < 0.001) | 7.27 (11.5,<0.001) | 11819 | 167.3 | 6.67 | 7034 | 165.0 | 7.22 | 718 | 158.5 | 5.58 |
| 1980–1994 | 23.4 | 3.3 | 19.5–34.3 | 14694 | 164.7 (118, < 0.001) | 7.07 (6.2,<0.001) | 7291 | 165.6 | 6.77 | 6274 | 164.9 | 6.96 | 633 | 159.8 | 5.74 |

[*]Welch ANOVA test for equality of means

[†]Levene's test for equality of variances; SD: standard deviation

twin individual including measurement error was more similar across birth-year cohorts. Heritability estimates ranged from 0.69 to 0.84 and were greatest in the two earliest and the two latest birth-year cohorts (*Table 2*). In women, although the total variance also started to increase from birth cohort 1940–1949, genetic variance showed an increasing trend from the earliest birth-year cohort. Both shared and unique environmental factors explained variation in height in all analyzed birth-year cohorts; whereas the shared environmental variance was somewhat greater in the latest cohorts (1970–1979 and 1980–1994) unique environmental variance was greatest in the earliest one. Although the variance components differed between sexes in all birth-year cohorts, the relative contribution of the genetic and environmental variance components did not differ by sex from 1930–1939 to 1960–1969 (*Supplementary file1B*). In contrast to the results in men, heritability estimates in women (0.53 to 0.78) were lowest in the earliest and latest cohorts, particularly in 1886–1909.

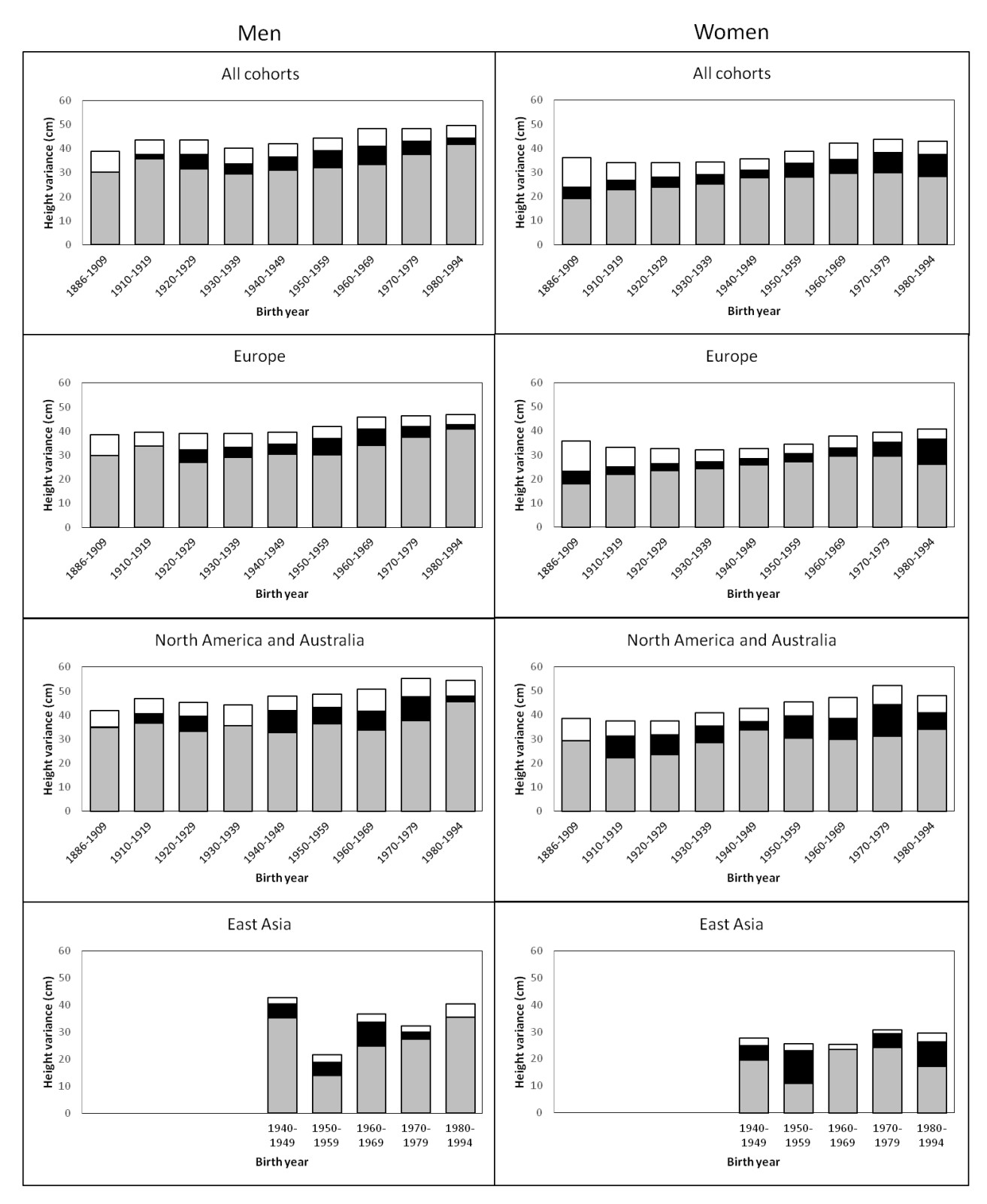

**Figure 1.** Additive genetic (grey), shared environmental (black) and unique environmental (white) variances of height across birth-year cohorts for the pooled data and by geographic-cultural region.

**Table 2.** Proportion of the height variance explained by additive genetic, shared environmental and unique environmental factors by birth year, sex and geographic-cultural region.

| | Men | | | | | | | | | Women | | | | | | | | |
|---|---|---|---|---|---|---|---|---|---|---|---|---|---|---|---|---|---|---|
| | Additive genetics | | | Shared environment | | | Unique environment | | | Additive genetics | | | Shared environment | | | Unique environment | | |
| Birth year | A | 95% CIs | | C | 95% CIs | | E | 95% CIs | | A | 95% CIs | | C | 95% CIs | | E | 95% CIs | |
| **All cohorts** | | | | | | | | | | | | | | | | | | |
| 1886–1909 | 0.78 | 0.69 | 0.80 | 0.00 | 0.00 | 0.08 | 0.22 | 0.20 | 0.25 | 0.53 | 0.43 | 0.62 | 0.13 | 0.05 | 0.21 | 0.34 | 0.31 | 0.37 |
| 1910–1919 | 0.82 | 0.76 | 0.87 | 0.04 | 0.00 | 0.10 | 0.14 | 0.13 | 0.15 | 0.67 | 0.60 | 0.74 | 0.11 | 0.04 | 0.18 | 0.22 | 0.20 | 0.24 |
| 1920–1929 | 0.72 | 0.69 | 0.76 | 0.14 | 0.10 | 0.17 | 0.14 | 0.13 | 0.15 | 0.70 | 0.64 | 0.76 | 0.12 | 0.07 | 0.18 | 0.18 | 0.17 | 0.19 |
| 1930–1939 | 0.73 | 0.68 | 0.79 | 0.10 | 0.04 | 0.16 | 0.16 | 0.15 | 0.18 | 0.74 | 0.68 | 0.79 | 0.11 | 0.06 | 0.16 | 0.15 | 0.14 | 0.16 |
| 1940–1949 | 0.74 | 0.70 | 0.78 | 0.13 | 0.09 | 0.17 | 0.13 | 0.12 | 0.13 | 0.78 | 0.75 | 0.82 | 0.09 | 0.05 | 0.13 | 0.13 | 0.12 | 0.13 |
| 1950–1959 | 0.72 | 0.69 | 0.76 | 0.16 | 0.12 | 0.20 | 0.12 | 0.11 | 0.12 | 0.73 | 0.69 | 0.76 | 0.15 | 0.11 | 0.18 | 0.12 | 0.12 | 0.13 |
| 1960–1969 | 0.69 | 0.63 | 0.76 | 0.16 | 0.09 | 0.21 | 0.15 | 0.14 | 0.16 | 0.70 | 0.66 | 0.75 | 0.14 | 0.09 | 0.18 | 0.16 | 0.15 | 0.17 |
| 1970–1979 | 0.77 | 0.72 | 0.83 | 0.11 | 0.06 | 0.17 | 0.11 | 0.10 | 0.12 | 0.68 | 0.64 | 0.73 | 0.19 | 0.14 | 0.23 | 0.13 | 0.12 | 0.13 |
| 1980–1994 | 0.84 | 0.77 | 0.90 | 0.05 | 0.00 | 0.13 | 0.11 | 0.10 | 0.12 | 0.66 | 0.61 | 0.72 | 0.21 | 0.16 | 0.27 | 0.13 | 0.12 | 0.13 |
| **Europe** | | | | | | | | | | | | | | | | | | |
| 1886–1909 | 0.78 | 0.69 | 0.80 | 0.00 | 0.00 | 0.08 | 0.22 | 0.20 | 0.25 | 0.50 | 0.40 | 0.60 | 0.14 | 0.06 | 0.23 | 0.35 | 0.32 | 0.39 |
| 1910–1919 | 0.85 | 0.79 | 0.87 | 0.00 | 0.00 | 0.07 | 0.15 | 0.13 | 0.16 | 0.66 | 0.58 | 0.74 | 0.10 | 0.02 | 0.17 | 0.24 | 0.22 | 0.26 |
| 1920–1929 | 0.69 | 0.62 | 0.76 | 0.14 | 0.07 | 0.20 | 0.17 | 0.16 | 0.19 | 0.72 | 0.65 | 0.79 | 0.09 | 0.03 | 0.16 | 0.19 | 0.17 | 0.21 |
| 1930–1939 | 0.75 | 0.69 | 0.81 | 0.11 | 0.05 | 0.17 | 0.14 | 0.13 | 0.16 | 0.76 | 0.70 | 0.82 | 0.09 | 0.03 | 0.14 | 0.16 | 0.15 | 0.17 |
| 1940–1949 | 0.77 | 0.72 | 0.82 | 0.10 | 0.06 | 0.15 | 0.13 | 0.12 | 0.13 | 0.79 | 0.75 | 0.83 | 0.08 | 0.04 | 0.13 | 0.13 | 0.12 | 0.13 |
| 1950–1959 | 0.72 | 0.68 | 0.77 | 0.16 | 0.11 | 0.20 | 0.12 | 0.11 | 0.12 | 0.79 | 0.75 | 0.83 | 0.09 | 0.05 | 0.13 | 0.12 | 0.11 | 0.13 |
| 1960–1969 | 0.74 | 0.66 | 0.83 | 0.15 | 0.06 | 0.23 | 0.11 | 0.10 | 0.12 | 0.78 | 0.72 | 0.85 | 0.08 | 0.02 | 0.15 | 0.13 | 0.12 | 0.14 |
| 1970–1979 | 0.81 | 0.74 | 0.88 | 0.09 | 0.02 | 0.16 | 0.10 | 0.09 | 0.10 | 0.74 | 0.69 | 0.81 | 0.15 | 0.09 | 0.21 | 0.11 | 0.10 | 0.11 |
| 1980–1994 | 0.87 | 0.77 | 0.92 | 0.04 | 0.00 | 0.14 | 0.09 | 0.08 | 0.10 | 0.64 | 0.57 | 0.72 | 0.26 | 0.18 | 0.32 | 0.10 | 0.09 | 0.11 |
| **North America and Australia** | | | | | | | | | | | | | | | | | | |
| 1886–1909 | 0.83 | 0.33 | 0.90 | 0.01 | 0.00 | 0.49 | 0.16 | 0.10 | 0.26 | 0.76 | 0.43 | 0.82 | 0.00 | 0.00 | 0.31 | 0.24 | 0.18 | 0.32 |
| 1910–1919 | 0.78 | 0.70 | 0.87 | 0.09 | 0.00 | 0.17 | 0.13 | 0.12 | 0.15 | 0.60 | 0.44 | 0.78 | 0.24 | 0.06 | 0.39 | 0.16 | 0.14 | 0.19 |
| 1920–1929 | 0.73 | 0.69 | 0.77 | 0.14 | 0.10 | 0.18 | 0.13 | 0.12 | 0.14 | 0.63 | 0.52 | 0.76 | 0.22 | 0.09 | 0.33 | 0.15 | 0.14 | 0.17 |
| 1930–1939 | 0.81 | 0.66 | 0.83 | 0.00 | 0.00 | 0.14 | 0.19 | 0.17 | 0.22 | 0.70 | 0.59 | 0.82 | 0.17 | 0.04 | 0.28 | 0.13 | 0.12 | 0.15 |
| 1940–1949 | 0.69 | 0.61 | 0.77 | 0.19 | 0.10 | 0.27 | 0.13 | 0.12 | 0.14 | 0.80 | 0.72 | 0.87 | 0.08 | 0.00 | 0.15 | 0.13 | 0.12 | 0.14 |
| 1950–1959 | 0.75 | 0.68 | 0.82 | 0.14 | 0.07 | 0.21 | 0.11 | 0.10 | 0.12 | 0.67 | 0.61 | 0.73 | 0.21 | 0.15 | 0.26 | 0.13 | 0.12 | 0.13 |
| 1960–1969 | 0.66 | 0.58 | 0.76 | 0.16 | 0.06 | 0.24 | 0.18 | 0.17 | 0.20 | 0.63 | 0.57 | 0.70 | 0.18 | 0.11 | 0.24 | 0.18 | 0.17 | 0.20 |
| 1970–1979 | 0.68 | 0.57 | 0.81 | 0.18 | 0.05 | 0.29 | 0.14 | 0.13 | 0.16 | 0.60 | 0.53 | 0.67 | 0.25 | 0.18 | 0.32 | 0.15 | 0.14 | 0.16 |
| 1980–1994 | 0.83 | 0.72 | 0.89 | 0.04 | 0.00 | 0.16 | 0.12 | 0.11 | 0.14 | 0.71 | 0.62 | 0.81 | 0.14 | 0.04 | 0.23 | 0.15 | 0.14 | 0.16 |
| **East Asia** | | | | | | | | | | | | | | | | | | |
| 1940–1949 | 0.83 | 0.33 | 0.97 | 0.12 | 0.00 | 0.61 | 0.05 | 0.03 | 0.12 | 0.71 | 0.17 | 0.94 | 0.19 | 0.00 | 0.73 | 0.10 | 0.06 | 0.18 |
| 1950–1959 | 0.64 | 0.24 | 0.91 | 0.23 | 0.00 | 0.63 | 0.13 | 0.08 | 0.20 | 0.42 | 0.14 | 0.92 | 0.48 | 0.00 | 0.75 | 0.10 | 0.07 | 0.15 |
| 1960–1969 | 0.67 | 0.36 | 0.94 | 0.24 | 0.00 | 0.56 | 0.08 | 0.06 | 0.12 | 0.92 | 0.67 | 0.94 | 0.00 | 0.00 | 0.25 | 0.08 | 0.06 | 0.10 |
| 1970–1979 | 0.85 | 0.51 | 0.95 | 0.08 | 0.00 | 0.43 | 0.07 | 0.05 | 0.09 | 0.79 | 0.52 | 0.96 | 0.17 | 0.00 | 0.43 | 0.05 | 0.04 | 0.06 |
| 1980–1994 | 0.88 | 0.51 | 0.91 | 0.00 | 0.00 | 0.37 | 0.12 | 0.09 | 0.17 | 0.58 | 0.34 | 0.90 | 0.31 | 0.00 | 0.55 | 0.11 | 0.09 | 0.14 |

When we studied the effect of birth year on the genetic variance by using gene-environment interaction models, modest but statistically significant increase was found. The interaction effect was 0.050 (95% CI 0.018–0.082) in men and 0.043 (95% CI 0.019–0.071) in women for the genetic path coefficient per 10 years. This turns to 1.37 (95% CI 0.50–2.27) increase of genetic variance in men and 1.07 (95% CI 0.46–1.79) increase of genetic variance in women per 25 years, i.e. approximately one human generation.

Univariate quantitative genetic models for height were then conducted separately in the three geographic-cultural regions (*Figure 1* and *Supplementary file 1A*). The pattern in Europe was practically the same as that observed for the pooled data because it represents a large proportion of the total sample. In North America and Australia, the total variance of height was greater than in Europe, but the pattern of genetic and environmental variances was less consistent across birth-year cohorts. In East Asia, because of the smaller sample size, the magnitude of the variance components between the birth-year cohorts fluctuated more than in the other two geographic-cultural regions. Genetic variance was generally greater in men than in women in the three geographic-cultural regions. Variance components of height (both raw and relative proportion) showed a similar pattern across birth-year cohorts when analyses were performed for men and women together (*Supplementary file 1C*).

## Discussion

This very large twin study showed no clear pattern in the heritability of height across birth-year cohorts and thus does not support the hypothesis that the heritability of height is lower in populations with low living standards compared with affluent populations, nor that the heritability of height will increase within a population as living standards improve. Since infant mortality rates are higher in men than in women, both in singletons (*Drevenstedt et al., 2008*) and twins (*Pongou, 2013*), the higher heritability observed for men in the earliest cohorts could be explained by selection effects since those who survived were the genetically more advantaged and thus less vulnerable to environmental conditions. The greater relative environmental effect on height variation in women than in men, although unexpected because women's growth is considered to be more resistant to environmental influences, is in agreement with the findings in Finnish twins born prior to 1958 (*Silventoinen et al., 2000*). This might indicate differential access to food and medical care (*Eveleth and Tanner, 1990*). Women are also more likely to develop osteoporosis leading to shrinking in old age (*National Institute of Arthritis and musculoskeletal and Skin Diseases, 2014*), which may affect the greater influence of unique environmental factors in women born in 1886–1910. This idea is supported by results showing that although genetic factors play an important role in bone loss in early postmenopausal women, their effect weakens with age and completely disappears with advanced aging (*Moayyeri et al., 2012*).

Total and genetic variance of height generally increased across birth-year cohorts; gene-birth year interaction analysis showed that the genetic variance increase was only modest even when it was statistically significant in this very large twin cohort. However, part of the increase in total variance in some birth-year cohorts was also due to the increase in shared environmental variance. This suggests that both greater ethnic diversity and variation in living standards have contributed to the secular increase in height variation. The greatest total height variation in North America and Australia was due to both genetic and environmental factors and the pattern of variance components across the birth cohorts was less consistent than in Europe. A recent study across 14 European countries found that many independent loci contribute to population genetic differences in height and estimated that these differences account for 24% of the captured additive genetic variance (*Robinson et al., 2015*). Therefore, it may be that both allelic frequencies and the effects of genes affecting height vary between the geographic-cultural regions. It has been previously shown that even when the total variance of height was greater in Western populations than in East Asian populations, heritability estimates were largely similar in adolescence (*Hur et al., 2008*) and from 1 to 19 years of age (*Jelenkovic et al., 2016*); however, the limited statistical power in the data from East Asia does not allow for comparisons across birth cohorts.

The main strength of the present study is the very large sample size of our multinational database of twin cohorts, with adult height data from individuals born between year 1886 and 1994, allowing a more detailed investigation of the genetic and environmental contributions to individual

differences in height across birth cohorts than in the previous studies. Important advantages of individual-based data are improved opportunities for statistical modeling and lack of publication bias. This type of analysis is difficult to perform by using literature-based meta-analyses because most of the published studies do not provide the needed statistics by birth-year cohorts. However, our study also has limitations. Countries and/or ethnic-cultural regions are not equally represented and the database is heavily weighted toward populations following Westernized lifestyles. In the classical twin design, parental phenotypic assortment increases dizygotic correlations and thus inflates the shared environmental component when not accounted for in the modeling. In our database, we do not have information on parental height and thus could not take into account assortative mating, which may thus explain part of the shared environmental variation. In addition, most of the height measures were self-reported (*Silventoinen et al., 2015*), which may bias our analyses toward higher estimates of unique environmental effects due to increased measurement error. However, these sources of bias are unlikely to explain our main result, i.e., relatively similar heritability estimates of adult height over birth cohorts. Finally, since we previously showed that there was no zygosity difference in height variance (*Jelenkovic et al., 2015*), variance components estimates should not be affected by changes in the proportion of MZ to DZ twins across birth-year cohorts.

In conclusion, although the genetic variance of height showed a slightly increasing trend with birth year, heritability estimates did not present any clear pattern of secular changes across birth-year cohorts from 1886 to 1994. Thus, our findings do not support the hypothesis that the heritability of height increases along with increasing living standards and diminishing rate of absolute poverty within populations.

## Materials and methods

### Sample

This study is based on the data from the Collaborative project of Development of Anthropometrical measures in Twins (CODATwins), which was intended to pool data from all twin projects in the world with information on height and weight measurements for MZ and DZ twins (*Silventoinen et al., 2015*). For the present analyses, we selected height measurements at ages 19.5–99.5 years. After excluding four cohorts having less than 50 twin individuals in the final database, we had data from 40 cohorts in 20 countries. The participating twin cohorts are identified in *Table 1* (footnote) and were previously described in detail (*Silventoinen et al., 2015*).

From the initial 558,672 height measurements, we excluded those <145 or>210 cm in men and <135 or >195 cm in women (<0.1% of the measurements). Since individuals in longitudinal studies have more than one measurement over time, analyses were restricted to one observation per individual resulting in 323,491 individuals. After excluding unmatched pairs (without data on their co-twins), we had 286,780 twin individuals (143,390 complete twin pairs) born between year 1886 and 1994 (40% monozygotic (MZ), 41% same- sex dizygotic (SSDZ) and 19% opposite-sex dizygotic (OSDZ) twin pairs). The smaller proportion of OSDZ compared to SSDZ twins in this study is explained by the fact that some of the twin cohorts in our database have collected, by design, only SSDZ twins and thus do not have data on OSDZ twins. These individuals were categorized into nine consecutive birth year groups described in *Table 1*. In order to analyze possible ethnic-cultural differences in the genetic and environmental contribution on height, cohorts were grouped in three geographical-cultural regions: Europe (18 cohorts), North America and Australia (14 cohorts) and East Asia (six cohorts) with 87,116, 53,359 and 1793 twin pairs, respectively. One cohort from the Middle-East and the one from South-Asia were not included in these sub-analyses by geographic-cultural region because the data were too sparse to study these two areas separately.

### Statistical analyses

We first tested whether the means and variances of height differed between twin cohorts within each sex and birth-year group (*Table 1*). Since the Levene´s test for homogeneity indicated that variances were not homogeneous, a Welch's ANOVA was performed showing that means were significantly different between twin cohorts in all sex and birth-year groups.

To analyze genetic and environmental influences on the variation of height, we used classic twin modeling based on linear structural equations (*Neale and Cardon, 1992*). MZ twins share the same

genomic sequence, whereas DZ twins share, on average, 50% of their genes identical-by-descent. On this basis, it is possible to divide the total variance of height into variance due to additive genetic effects (A: correlated 1.0 for MZ and 0.5 for DZ pairs), dominance genetic effects (D: 1.0 for MZ and 0.25 for DZ pairs), common (shared) environmental effects (C: by definition, correlated 1.0 for MZ and DZ pairs) and unique (non-shared) environmental effects (E: by definition, uncorrelated in MZ and DZ pairs). However, since our data included only twins reared together, we cannot simultaneously estimate shared environmental and dominance genetic effects. All genetic models were fitted by the OpenMx package (version 2.0.1) in the R statistical platform (*Boker et al., 2011*) using the maximum likelihood method.

Prior to conducting the modeling, height values were adjusted for the year of birth and twin cohort within each birth year and sex groups using linear regressions, and the resulting residuals were used as input phenotypes. The ACE sex-limitation model was selected as a starting point of the univariate modeling based on the following criteria: (i) MZ within-pair correlations were clearly higher than DZ correlations consistent with the influence of genetic effects, (ii) the magnitude of the difference between MZ and DZ correlations (rDZ > 1/2 rMZ) indicated the presence of common environmental effects and (iii) the lower within-pair correlations for OSDZ than for SSDZ twins observed for most birth-year groups suggested the presence of sex-specific genetic effects (results not shown). Previous findings from this international database showed that both male and female DZ twins are slightly taller than MZ twins in these age groups (*Jelenkovic et al., 2015*), and thus different means for MZ and DZ twins were allowed. The fit of the univariate models for height at each birth-year group is shown in *Supplementary file 1B*. In the present study, the equal-environment assumption was tested by comparing the ACE model to the saturated model. The fit of the models after Bonferroni correction of multiple testing did not worsen for most birth-year groups, which suggested that the assumption of equality of variances between MZ and DZ twins was not violated. When fixing A, C and E parameters to be the same in men and women, the fit of the model was poorer in all birth-year groups (p<0.0001), suggesting that these variance components differ between sexes. We additionally fitted a scale model allowing for different sizes of variance components but fixing the relative size of these components to be equal. Since this model also showed statistically significant differences (p<0.0001) in some birth-year cohorts, we decided to present the results separately for men and women. Sex-specific genetic effects were significant for some birth-year cohorts, and thus all modeling results are presented in sex-limited form for consistency. Comparative model fitting revealed that the C parameter could be not excluded from the model without a significant deterioration in fit. In order to study how birth year modifies the genetic and environmental variances of height, we additionally conducted gene-environment interaction modeling using birth year as an environmental modification factor (*Purcell, 2002*). This modeling offers intercept and interaction term describing the change per birth year which then need to be squared to get raw genetic and environmental variances. To make the results easier to understand, we calculated expected variance change with 95% CI per 25 years, i.e. approximately one human generation.

## Acknowledgements

Support for collaborating projects: The Australian Twin Registry is supported by a Centre of Research Excellence (grant ID 1079102) from the National Health and Medical Research Council administered by the University of Melbourne. The California Twin Program was supported by The California Tobacco-Related Disease Research Program (7RT-0134H, 8RT-0107H, 6RT-0354H) and the National Institutes of Health (1R01ESO15150-01). The Carolina African American Twin Study of Aging (CAATSA) was funded by a grant from the National Institute on Aging (grant 1RO1-AG13662-01A2) to K. E. Whitfield. Colorado Twin Registry is funded by NIDA funded center grant DA011015, and Longitudinal Twin Study HD10333; Author Huibregtse is supported by 5T32DA017637-11. Danish Twin Registry is supported by the National Program for Research Infrastructure 2007 from the Danish Agency for Science, Technology and Innovation, The Research Council for Health and Disease, the Velux Foundation and the US National Institute of Health (P01 AG08761). Since its origin the East Flanders Prospective Survey has been partly supported by grants from the Fund of Scientific Research, Flanders and Twins, a non-profit Association for Scientific Research in Multiple Births (Belgium). Data collection and analyses in Finnish twin cohorts have been supported by ENGAGE – European Network for Genetic and Genomic Epidemiology, FP7-HEALTH-F4-2007, grant agreement

number 201413, National Institute of Alcohol Abuse and Alcoholism (grants AA-12502, AA-00145, and AA-09203 to R J Rose, the Academy of Finland Center of Excellence in Complex Disease Genetics (grant numbers: 213506, 129680), and the Academy of Finland (grants 100499, 205585, 118555, 141054, 265240, 263278 and 264146 to J Kaprio). K Silventoinen is supported by Osaka University's International Joint Research Promotion Program. Waves 1–3 of Genesis 12–19 were funded by the W T Grant Foundation, the University of London Central Research fund and a Medical Research Council Training Fellowship (G81/343) and Career Development Award (G120/635) to Thalia C. Eley. Wave four was supported by grants from the Economic and Social Research Council (RES-000-22–2206) and the Institute of Social Psychiatry (06/07–11) to Alice M. Gregory who was also supported at that time by a Leverhulme Research Fellowship (RF/2/RFG/2008/0145). Wave five was supported by funding to Alice M. Gregory from Goldsmiths, University of London. Anthropometric measurements of the Hungarian twins were supported by Medexpert Ltd., Budapest, Hungary. Korean Twin-Family Register was supported by the Global Research Network Program of the National Research Foundation (NRF 2011–220-E00006). The Michigan State University Twin Registry has been supported by Michigan State University, as well as grants R01-MH081813, R01-MH0820-54, R01-MH092377-02, R21-MH070542-01, R03-MH63851-01 from the National Institute of Mental Health (NIMH), R01-HD066040 from the Eunice Kennedy Shriver National Institute for Child Health and Human Development (NICHD), and 11-SPG-2518 from the MSU Foundation. The content of this manuscript is solely the responsibility of the authors and does not necessarily represent the official views of the NIMH, the NICHD, or the National Institutes of Health. *The Murcia Twin Registry is supported by Fundación Séneca, Regional Agency for Science and Technology, Murcia, Spain (08633/PHCS/08, 15302/PHCS/10 and 19479/PI/14) and Ministry of Science and Innovation, Spain (PSI2009-11560 and PSI2014-56680-R).* Data collection and research stemming from the Norwegian Twin Registry is supported, in part, from the European Union's Seventh Framework Programmes ENGAGE Consortium (grant agreement HEALTH-F4-2007–201413, and BioSHaRE EU (grant agreement HEALTH-F4-2010–261433). The NAS-NRC Twin Registry acknowledges financial support from the National Institutes of Health grant number R21 AG039572. Netherlands Twin Register acknowledges the Netherlands Organization for Scientific Research (NWO) and MagW/ZonMW grants 904-61–090, 985-10–002, 912-10–020, 904-61–193,480-04–004, 463-06–001, 451-04–034, 400-05–717, Addiction-31160008, Middelgroot-911-09–032, Spinozapremie 56-464–14192; VU University's Institute for Health and Care Research (EMGO+); the European Research Council (ERC - 230374), the Avera Institute, Sioux Falls, South Dakota (USA). South Korea Twin Registry is supported by the National Research Foundation of Korea (NRF-371-2011–1 B00047). S.Y. Öncel and F. Aliev are supported by Kırıkkale University Research Grant: KKU, 2009/43 and TUBITAK grant 114C117. TwinsUK was funded by the Wellcome Trust; European Community's Seventh Framework Programme (FP7/2007–2013). The study also receives support from the National Institute for Health Research (NIHR) BioResource Clinical Research Facility and Biomedical Research Centre based at Guy's and St Thomas' NHS Foundation Trust and King's College London. The University of Southern California Twin Study is funded by a grant from the National Institute of Mental Health (R01 MH58354). Washington State Twin Registry (formerly the University of Washington Twin Registry) was supported in part by grant NIH RC2 HL103416 (D. Buchwald, PI). Vietnam Era Twin Study of Aging was supported by the National Institute of Health grants NIA R01 AG018384, R01 AG018386, R01 AG022381, and R01 AG022982, and, in part, with resources of the VA San Diego Center of Excellence for Stress and Mental Health. The Cooperative Studies Program of the Office of Research and Development of the United States Department of Veterans Affairs has provided financial support for the development and maintenance of the Vietnam Era Twin (VET) Registry. The content of this manuscript is solely the responsibility of the authors and does not necessarily represent the official views of the NIA/NIH, or the VA. The West Japan Twins and Higher Order Multiple Births Registry was supported by Grant-in-Aid for Scientific Research (B) (grant number 15H05105) from the Japan Society for the Promotion of Science.

## Additional information

### Funding

| Funder | Grant reference number | Author |
|--------|------------------------|--------|
| Suomen Akatemia | 266592 | Karri Silventoinen |

The funders had no role in study design, data collection and interpretation, or the decision to submit the work for publication.

### Author contributions

AJ, In charge of data management, Conducted the analyses, Wrote the first draft of the manuscript and has primary responsibility of the final content; Y-MH, YY, KOK, FRa, DIB, TIAS, JK, KS, Planned the study design of the CODATwins project, Collected the data used in this study, Commented the manuscript, Read and approved the final version of the manuscript; RS, SHS, MH, ASu, FRi, QT, DZ, ZP, SA, KH, SYÖ, FA, ER, ADT, DLT, KC, ASk, JLS, LJE, HHM, TLC, JLH, JRO, JFS-R, LC-C, WC, AEH, TMM, JS, Y-MS, SY, KL, CEF, WSK, MJL, AB, TLN, KEW, CK, KLJ, MG, DAB, MAS, CF, CD, GED, DB, CAD, RFV, RJFL, NGM, SEM, GWM, H-UJ, GES, RK, PKEM, NLP, AKD-A, TAM, TCE, AMG, PT, LAB, CT, GB, DN, PL, TDS, MMa, GL, MB, TCEMvB, GW, SAB, KLK, JRH, IB, TSN, RFK, MMcGu, SP, RPC, JvBH, JHG, YI, MW, CH, FI, BMH, Collected the data used in this study, Commented the manuscript, Read and approved the final version of the manuscript

### Ethics

Human subjects: All participants were volunteers and gave their informed consent when participating in their original study. Only a limited set of observational variables and anonymized data were delivered to the data management center at University of Helsinki. The pooled analysis was approved by the ethical committee of Department of Public Health, University of Helsinki.

## Additional files

### Supplementary files

• Supplementary file 1. Supplementary tables. (A) Height variance explained by additive genetic, shared environmental and unique environmental factors by birth year, sex and geographic-cultural region. (B) Model fit statistics for adult height by birth-year cohorts (all twin cohorts together). (C) Height variance and proportion of height variance explained by additive genetic, shared environmental and unique environmental factors by birth year, sex and geographic-cultural region.

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
