## [Decision Letter]

Thank you for submitting your article "Genetic and environmental influences on adult human height across birth cohorts from 1886 to 1994" for consideration by *eLife*. Your article has been favorably evaluated by Prabhat Jha (Senior Editor) and four reviewers, one of whom, Eduardo Franco, is a member of our Board of Reviewing Editors. The following individual involved in review of your submission has agreed to reveal their identity: Timothy Frayling.

The reviewers have discussed the reviews with one another and the Reviewing Editor has drafted this decision to help you prepare a revised submission.

Summary:

This study tested the plausible hypothesis – grounded on economic theory – that the genetic influences on adult height may have increased over the last several decades as living standards improved, which would have decreased the environmental constraints from socioeconomic deprivation on attained height. The investigators could only test this hypothesis because of the opportunity to pool together via a massive consortium many twin cohorts across multiple continents. They concluded that the hypothesis could not be confirmed, as the heritability of height did not follow any clear secular trends, irrespective of population. This study addresses an interesting question using the power of almost all, if not all, the twin studies in the world – how has the genetic and environmental component to height altered over the last 100 years? Given all the studies included this can be regarded as as close to the definitive study as one can get.

Essential revisions:

The following are specific points summarized from the reviewers' critiques. They require your attention for us to consider a revised version of your paper.

1) Could the relative proportion of MZ to DZ twins have changed over time and this affected the conclusions of this study? The entire set included 40% MZ, 41% same-sex DZ, and 19% opposite-sex DZ. Greater than average height and weight increases the likelihood of a pregnancy resulting in DZ twins. Age over 40 and prior history does the same. Average age at first pregnancy and numbers thereof are likely to have changed over time and represent variables that could be sensitive to era effects related to war and other determinants of deprivation. On the other hand, there are no known genetic correlates of MZ pregnancies and the rate is constant across populations. Another potential confounder related to the above is the improvements in obstetric care over time, which would have increased the survivability of MZ twins.

2) As a sidebar to the above question: Shouldn't the proportion of opposite-sex DZ be higher than what the study found?

3) I am intrigued by the apparent secular decreases in the proportion of variation due to the unique environment component. This happened for both men and women and were more noticeable for the European cohorts. It seems to me that this suggests that the original declining-deprivation hypothesis has some merit. The authors focused on the additive genetic component but did not discuss much what happened to the other components.

4) It is worth considering presenting the results of both sexes together as well as split by sex. Whilst I can think of reasons why changes in the heritability of height may differ by sex, it is not clear why the authors have stratified their primary analysis by sex. If the main hypothesis is that secular changes will change the heritability of height, one would expect these to operate in the childhood growth of boys and girls. Perhaps girls will be less susceptible because they grow for a shorter period of their lives. But why reduce the power of the study by half? (Especially when there are wider confidence intervals around estimates from before 1940s.)

5) Are the differences between men and women in the earliest time points significant? It is not clear. The authors speculate that there may have been stronger survival effects in men, but this will be unnecessary speculation if there is no evidence of a difference between sexes.

6) It is not possible to prove the negative. Instead can the authors place some bounds on their conclusions? E.g., "we could exclude an increase of x% genetic variance per generation (25 years) with 95% confidence"? On a related note, there are no p values or effect sizes anywhere in the main text. This makes the reader take the results on faith (e.g. It is not clear whether "trend" means a statistically robust trend or just a hint). I realise these appear in the supplementary information, but I think it would help the reader to see the genetic variances across time with 95% CIs at least (and combined sexes would be most powerful).

7) Can the authors comment more on the overall increase in variance observed? It is worth noting that the genetic variance appears to go up in line with the overall variance. The reasons for this are not testable I imagine, but presumably could be due to increased ethnic diversity, and greater variation in living standards, as the average increases. It is clear in Figure 1 but not in the text.

8) [Supplementary-material SD1-data]: Table 1 and Table 2 should be part of the main article. It would help the reader to see some stats in the main section of the paper.

---

## [Author Response]

**[…]**

*Essential revisions:*

*The following are specific points summarized from the reviewers' critiques. They require your attention for us to consider a revised version of your paper.*

*1) Could the relative proportion of MZ to DZ twins have changed over time and this affected the conclusions of this study? The entire set included 40% MZ, 41% same-sex DZ, and 19% opposite-sex DZ. Greater than average height and weight increases the likelihood of a pregnancy resulting in DZ twins. Age over 40 and prior history does the same. Average age at first pregnancy and numbers thereof are likely to have changed over time and represent variables that could be sensitive to era effects related to war and other determinants of deprivation. On the other hand, there are no known genetic correlates of MZ pregnancies and the rate is constant across populations. Another potential confounder related to the above is the improvements in obstetric care over time, which would have increased the survivability of MZ twins.*

As mentioned by the reviewer, changes in twinning rates are largely attributable to dizygotic (DZ) twinning. Monozygotic (MZ) twinning is considered an essentially random event with fairly constant rates worldwide, but a significant increase from 1960 has been reported for some countries (Imaizumi et al., 2003). This increasing MZ twinning rate could be explained by the improvements in obstetric care over time increasing the survivability of MZ twins but it has also been associated with increasing use of oral contraceptives (Imaizumi et al., 2003). in vitro fertilization (IVF) also causes MZ twinning occasionally (Aston et al., 2008). Changes in DZ twinning rates are influenced by maternal age, ethnicity, family history, and height and weight. The higher DZ twinning rate since the 1980s have been attributed to the widespread use of IVF and other fertility treatments in most industrialized countries (Imaizumi et al., 2003; Blickstein et al., 2005). Therefore, after the introduction of fertility drugs and IVF, variations in the DZ twinning were not only due to biological factors, but also depended on the popularity of fertility drugs and IVF in each country.

In this sample, the proportion of MZ to DZ twins across the studied birth-year cohorts (from 1886-1909 to 1980-1994) is as follows: 37%, 39%, 41%, 35%, 33%, 38%, 43%, 48%, 50%. This shows that the proportion of MZ to DZ twins is quite similar from 1886-1909 to 1950-1959 (33-41%) and starts to increase from 1960, which does not reflect the rise in DZ twins seen in developed countries during the past three decades. In a previous study on this database, we showed that there was no zygosity difference in height variance, neither in childhood nor in adulthood (Jelenkovic et al., 2015). Therefore, there is no reason to think that changes in the proportion of MZ to DZ twins would affect variance components estimates. This has now been discussed in limitations.

*2) As a sidebar to the above question: Shouldn't the proportion of opposite-sex DZ be higher than what the study found?*

The reviewer is right in that the proportion of same sex (SSDZ) and opposite sex (OSDZ) dizygotic twins should be the same. The considerably smaller proportion of OSDZ compared to SSDZ twins in this study is explained by the fact that some of the twin cohorts in our database have collected, by design, only SSDZ twins and thus do not have data on OSDZ twins. This has now been mentioned in the manuscript.

*3) I am intrigued by the apparent secular decreases in the proportion of variation due to the unique environment component. This happened for both men and women and were more noticeable for the European cohorts. It seems to me that this suggests that the original declining-deprivation hypothesis has some merit. The authors focused on the additive genetic component but did not discuss much what happened to the other components.*

A decreasing trend in the proportion of variation due to unique environmental factors (E) across birth-year cohorts was observed only for the four earliest birth cohorts, and was more noticeable in Europe and in women. That is, this trend was not observed in East Asia or North America and Australia (except for the slightly greater relative E variance in 1886-1909 for women in North America and Australia), nor in Europe from 1940 onwards. Moreover, in men, the decrease in relative E variance was not associated with a parallel increase in relative A variance (because relative C variance increased), which does not support the declining-deprivation hypothesis. That is, since height is influenced by environmental factors during the whole growth period (particularly in infancy and puberty), we expect that some of these environmental factors are shared by co-twins; therefore, and according to the hypothesis, this should have been seen as a decrease in C variance, which was not observed. In fact, in several cases a decrease in relative E variance was associated with an increase in relative C variance. If we look at the raw variances, the decreasing trend in E variance in the earliest birth cohorts is noticeable for women but not clear for men.

In summary, the parallel decrease in relative E variance and increase in relative A variance was observed only in European women for the four earliest birth-year cohorts; in fact, the heritability estimate decreased again in the two latest birth cohorts. Therefore, we alternatively speculated that the *greater influence of unique environmental factors in the earliest birth cohorts in women*might be explained by*shrinkage in old age.* Finally, since shared environmental factors did not show any pattern across birth-years cohorts, we described the results but not discussed them in the Discussion.

*4) It is worth considering presenting the results of both sexes together as well as split by sex. Whilst I can think of reasons why changes in the heritability of height may differ by sex, it is not clear why the authors have stratified their primary analysis by sex. If the main hypothesis is that secular changes will change the heritability of height, one would expect these to operate in the childhood growth of boys and girls. Perhaps girls will be less susceptible because they grow for a shorter period of their lives. But why reduce the power of the study by half? (Especially when there are wider confidence intervals around estimates from before 1940s.)*

We presented the results separately in men and women because the model fit statistics showed that the variance components differed between sexes in all birth-year cohorts, and the relative contribution of the genetic and environmental variance components differ in the three earliest and two latest birth-year cohorts (Table 2 in [Supplementary-material SD1-data]).

*As suggested by the reviewer, we have now estimated both raw and relative genetic and environmental variances for men and women together. However, we decided to present these combined results as a supplementary table (Table 3 in [Supplementary-material SD1-data]) because 1) they did not provide any additional information on the trend across birth-year cohorts compared to the results for men and women separately and 2) since variance components differed between sexes, we think it is more appropriate to estimate them separately in men and women. This has now been mentioned in the text.*

*5) Are the differences between men and women in the earliest time points significant? It is not clear. The authors speculate that there may have been stronger survival effects in men, but this will be unnecessary speculation if there is no evidence of a difference between sexes.*

*The variance components differed between sexes in all birth-year cohorts, and the relative contribution of the genetic and environmental variance components differed in the three earliest and two latest birth-year cohorts (*Table 2 *in [Supplementary-material SD1-data]). We have now mentioned in Methods section that these differences were statistically significant at p<0.0001. Based on these results, we think that it is worth to speculate that there may have been stronger survival effects in men.*

*6) It is not possible to prove the negative. Instead can the authors place some bounds on their conclusions? E.g., "we could exclude an increase of x% genetic variance per generation (25 years) with 95% confidence"? On a related note, there are no p values or effect sizes anywhere in the main text. This makes the reader take the results on faith (e.g. It is not clear whether "trend" means a statistically robust trend or just a hint). I realise these appear in the supplementary information, but I think it would help the reader to see the genetic variances across time with 95% CIs at least (and combined sexes would be most powerful).*

As suggested by the reviewer, we have quantified the increase in genetic variance per generation by using G-E interaction analyses. The results showed that genetic variance increased 1.37 (95% CI 0.50-2.27) and 1.07 (95% CI 0.46-1.79) per 25 years in men and women, respectively, which information is now given in the main text. The 95% CIs thus shows that the increase of genetic variance is statistically significant but the increasing effect in variance is still quite modest. As suggested by the reviewer, we have now also included in the main text the table showing the proportion of height variance explained by A, C and E factors. As previously explained in comment 4, we finally decided to present in the main text the results separately in men and women (and combined results in supplementary table) because variance components differed between sexes and thus we think that, even if less powerful, results are more correct.

*7) Can the authors comment more on the overall increase in variance observed? It is worth noting that the genetic variance appears to go up in line with the overall variance. The reasons for this are not testable I imagine, but presumably could be due to increased ethnic diversity, and greater variation in living standards, as the average increases. It is clear in Figure 1 but not in the text.*

Although there is a general trend to increasing total and genetic variance across birth cohorts, genetic variance does not always go up with total variance. For example, in men, the greatest increase in total variance was observed from birth cohort 1940-1949 to 1960-1969 and although genetic variance also increased during this period it increased especially in the two latest birth-year cohorts (1970-1979 and 1980-1994). In women, although total variance also started to increase from birth cohort 1940-1949, genetic variance showed the greatest increase from 1886-1900 to 1940-1949. As can be seen in Figure 1, part of the increase in total variance is due to the increase in shared environmental variance. Therefore, and as suggested by the reviewer, the increase in total height variation could be due to both increased ethnic diversity and greater variation in living standards. This has now been discussed in the text.

*8) [Supplementary-material SD1-data]: Table 1 and Table 2 should be part of the main article. It would help the reader to see some stats in the main section of the paper.*

Supplementary Table 2 is now part the main text as Table 2: however, we have not included Supplementary Table 1 because the results are already provided in Figure 1 (without CIs) and we think it would provide repeated information.